Retrospective analysis of cervical screening abnormalities in women with type 3 transformation zone without visible lesions

Yang Jing 1
Liu Qiao 1
Tang Yi 1
Huang Kui 1
Chen Tianmin 1
Zhao Jing jxhzj2005@hotmail.com 1
Department of Obstetrics & Gynecology, Hunan Provincial Maternal and Child Health Care Hospital , Changsha , Hunan , China
Anson Lesley
Electronic publication date: 2025 Nov 27
Publication date: 2025
Volume: 13
Electronic Location ID: e20396
Received 2025 Apr 21; Accepted 2025 Oct 24
Copyright: ©2025 Yang et al.
Copyright year: 2025
Copyright holder: Yang et al.
License: This is an open access article distributed under the terms of the Creative Commons Attribution License, which permits unrestricted use, distribution, reproduction and adaptation in any medium and for any purpose provided that it is properly attributed. For attribution, the original author(s), title, publication source (PeerJ) and either DOI or URL of the article must be cited.
License URL: https://creativecommons.org/licenses/by/4.0/

Keywords: Cervical cancer screening, Colposcopy, HSIL+, HPV 16/18, Endocervical curettage, Risk stratification, Age-related risk

Funding: Hunan Province Science and Technology Innovation Platform and Talents Program—Hunan Province Cervical Cancer Prevention and Clinical Research Center Fund 2021SK4021 Hunan Provincial Maternal and Child Health Care Hospital’s 2023 Sharp New Cultivation Program Projects 2023RX27 Natural Science Foundation of Hunan Province Medical and Health Industry Joint Fund Project 2024JJ9335 This study was supported by the Hunan Province Science and Technology Innovation Platform and Talents Program—Hunan Province Cervical Cancer Prevention and Clinical Research Center Fund (2021SK4021); Hunan Provincial Maternal and Child Health Care Hospital’s 2023 Sharp New Cultivation Program Projects (2023RX27); and Natural Science Foundation of Hunan Province Medical and Health Industry Joint Fund Project (2024JJ9335). The funders had no role in study design, data collection and analysis, decision to publish, or preparation of the manuscript.

==============================
Objective

Women with abnormal cervical screening but without visible lesions, particularly those with a type 3 transformation zone (TZ3), present a clinical challenge due to the non-visible squamocolumnar junction, increasing the risk of missed high-grade lesions. There is currently no consensus on optimal follow-up strategies for this group. This study aims to evaluate a risk-based management approach for these patients.

Methods

A cross-sectional study analyzed data from 4,648 women with TZ3 who underwent colposcopy and endocervical curettage (ECC) with or without cervical biopsies at Hunan Provincial Maternal and Child Health Care Hospital (2021–2024). Logistic regression with restricted cubic splines analyzed demographic, cytological and HPV data to identify HSIL+ predictors and age-risk thresholds.

Results

Among the study population, 3.1% (145 cases) of HSIL+ were identified despite negative colposcopy, although additional undetected cases may exist. Women with high-grade cytology (ASC-H/HSIL/AGC) had a consistently high HSIL+ risk (32.5%–37.2%) across all HPV subgroups. In low-grade cytology (NILM/ASCUS/LSIL), HPV 16/18 positivity increased HSIL+ risk (2.4%–5.0%) compared to non-HPV 16/18 cases (1.6%–1.8%), with the highest rate observed in LSIL cases (5.0%). In women with low-grade cytology and non-HPV 16/18 positivity, age and HSIL+ risk showed a nonlinear relationship (RCS P-nonlinear = 0.008). Threshold analysis identified 55 years as a critical cutoff, with a 10% annual increase in HSIL+ risk for women ≥ 55. (OR = 1.10, 95% CI [1.02–1.19]; P = 0.015). Further age-stratified analysis in this subgroup showed a clear upward trend: HSIL+ detection rates were 4.42% in women aged ≥ 65.

Conclusion

Among women with abnormal cervical screening and no visible lesions at type 3 transformation zone, HSIL+ risk varies by cytology, HPV genotype, and age. Our findings suggest that immediate diagnostic evaluation is warranted for those with high-grade cytology, HPV 16/18 with LSIL, and women aged ≥ 65 years with low-grade cytology and non-16/18 HPV, as their HSIL+ risk exceeds the 4% threshold recommended by current US guidelines. Conversely, women under 65 with low-grade cytology and non-16/18 HPV, or those with NILM/ASCUS and HPV 16/18, may be appropriate candidates for conservative follow-up. These results support a more tailored, risk-based approach to management in this challenging population.

Introduction

Cervical cancer remains a significant global health burden. In 2020, there were 604,127 new cases and 341,831 deaths reported worldwide (Singh et al., 2023). While organized screening programs utilizing cervical cytology and high-risk human papillomavirus (HPV) testing have substantially reduced cervical cancer incidence and mortality, challenges persist in managing women with screening abnormalities but inconclusive follow-up evaluations. In particular, cases with abnormal cytologies or HPV-positive results but without visible lesions represent a critical clinical dilemma, as these women may harbor occult high-grade squamous intraepithelial lesions (HSIL+) or face delayed diagnoses of precancerous lesions. Even among experienced colposcopists, the sensitivity of colposcopy for detecting cervical intraepithelial neoplasia (CIN) ranges from 81.4% to 95.7%, with specificity as low as 34.2% to 69% (Brown & Tidy, 2019; Underwood et al., 2012; Sideri et al., 2015; Massad et al., 2009; Huh, Papagiannakis & Gold, 2019). This variability underscores the critical need to minimize missed diagnoses, particularly high-grade lesions, in women with discordant screening and colposcopic results.

Current guidelines recommend colposcopy-guided biopsies for women with abnormal screening results (Perkins et al., 2020; Kawaguchi et al., 2019). However, for women with an invisible squamocolumnar junction (SCJ) and no visible lesions, the examination cannot reliably assess potential disease hidden within the endocervical canal, the necessity of endocervical curettage (ECC) or biopsies remains uncertain. According to the 2017 ASCCP Colposcopy Standards (Waxman et al., 2017) ECC is specifically recommended when the transformation zone is not fully visible (TZ3). This consensus recommendation is supported by studies demonstrating ECC significantly improves detection of HSIL+ in women with an invisible SCJ (Massad et al., 2023; Behrens et al., 2024; Liu et al., 2017), the greatest controversy lies in whether women with a positive screening result but a normal-appearing cervix on colposcopy should undergo random biopsy or not, or whether we should combine and use random biopsies and ECC together or not.

In China, primary cervical cancer screening typically involves co-testing with high-risk HPV testing and cytology. According to national screening guidelines, women are referred for colposcopy if either test yields abnormal results (Massad et al., 2013). HPV genotyping is commonly used to guide risk stratification, with HPV16/18-positive women referred directly to colposcopy, while other types may be triaged based on cytology. This dual-testing strategy is widely implemented in urban areas. However, management decisions become complex when colposcopic evaluation reveals no visible lesions, particularly in patients with an invisible squamocolumnar junction, highlighting the need for evidence-based follow-up strategies in this group.

Despite the effectiveness of cervical cancer screening programs, the best way to manage women with abnormal cytology or high-risk HPV infection but no visible colposcopic findings remains controversial. This study aims to address these gaps by analyzing a large dataset of women with type 3 transformation zone who had abnormal cervical screening results but no visible colposcopic findings.We assess the immediate risk of detecting HSIL+ and identify key predictors, including age, HPV type, and cytology. Additionally, we use advanced statistical modeling (piecewise logistic regression) to explore how risk factors affect outcomes and define thresholds for increasing surveillance. By understanding how demographic, virologic, and cytologic factors interact, this study provides practical insights for risk-stratified management. For high-risk groups, we further evaluate the diagnostic value of biopsies to improve detection accuracy. These findings enable clinicians to tailor follow-up strategies, optimize resource allocation, and minimize missed diagnoses.

Materials and Methods

Study design and population

This study utilized data extracted from the electronic medical records of women who underwent colposcopic evaluations at Hunan Provincial Maternal and Child Health Care Hospital, after receiving abnormal cervical screening results. Women eligible for inclusion were those with a history of sexual activity, not currently pregnant, with a structurally intact uterus, and no prior history of cervical malignancy or pelvic radiotherapy. Baseline demographic and clinical information included age, gravidity, parity, menopausal status, cervical cytology results, HPV status, colposcopic impression, type of transformation zone, and histopathological findings. This study was approved by the Ethics Committee of Hunan Provincial Maternal and Child Health Care Hospital (2023-S180), and the requirement for informed consent was waived due to its retrospective design. All patient data were fully anonymized prior to analysis, and no identifiable personal information was used. Data security and confidentiality were strictly maintained throughout the study.The findings were reported in accordance with the Strengthening the Reporting of Observational Studies in Epidemiology (STROBE) guidelines (Von Elm et al., 2007).

Cytology and HPV testing

Liquid-based cytology and HPV testing were performed for primary screening.The cytology screening test was conducted using a liquid-based method, with results categorized according to the 2001 Bethesda System (Solomon et al., 2002). Cytological abnormalities were defined to include atypical squamous cells of undetermined significance (ASC-US), low-grade squamous intraepithelial lesion (LSIL), high-grade squamous intraepithelial lesion (HSIL), atypical squamous cells—cannot rule out HSIL (ASC-H), atypical glandular cells (AGC), adenocarcinoma in situ (AIS), squamous cell carcinoma (SCC), and invasive adenocarcinoma. HPV genotyping was performed using cervical specimens collected at enrollment through PCR-based multicolor fluorescence assay. The interval between cytology/HPV testing and colposcopy was consistently less than 3 months.

Colposcopy, ECC, and biopsy procedures

Females with any positive primary screening were referred to colposcopy. Colposcopic examinations were performed using a Leisegang digital photoelectric colposcope (Germany). A total of 5% acetic acid was applied to the surface of the cervix and aceto-white changes on the cervix were recorded; Lugol’s iodine was applied in selected cases to assist lesion delineation, especially when acetowhitening was absent or equivocal, or when biopsy guidance was needed. According to the 2011 Colposcopic Terminology established by the International Federation for Cervical Pathology and Colposcopy (Bornstein et al., 2012), type 1 transformation zone (TZ1) is confined to the ectocervical surface. TZ 2 extends partially into the endocervical canal but can be completely visualized with appropriate techniques to dilate or expose the canal. TZ3 involves partial or complete extension into the endocervical canal and cannot be fully visualized in its entirety. This study focused on TZ3 cases, where no visible colposcopic findings were defined as the absence of acetowhitening, metaplasia, or other abnormalities. Histopathological results from both ECC and cervical biopsy specimens were classified using the Lower Anogenital Squamous Terminology (LAST) system (Cree et al., 2020), findings were categorized as normal, LSIL, HSIL, or invasive carcinoma. The final diagnostic outcome was based on the most advanced lesion identified, with HSIL+ defined to include HSIL, AIS, and invasive malignancies; all remaining results were considered <HSIL. Evaluation of ECC and biopsy slides was independently performed by two experienced pathologists, each blinded to the other’s assessment. Discrepant cases were reviewed jointly to reach consensus. If no agreement was achieved, a third senior pathologist rendered the final diagnosis.This study included only women with a positive screening result but no visible colposcopic findings who underwent ECC for analysis, most of whom also had random biopsies performed concurrently. ECC was performed first using a Kevorkian curette, followed by possible further biopsy using Tischler biopsy forceps. One to four random biopsies were obtained per patient from different quadrants around the external os.

For women with high-grade cytology (HSIL or ASC-H) but no visible colposcopic lesions and negative biopsies, cytologic results were first reviewed with senior cytopathologists, and slides were re-examined when necessary. Cases with persistent cytologic abnormalities and unsatisfactory colposcopy. Diagnostic excision, typically via loop electrosurgical excision procedure (LEEP), was recommended in accordance with national guidelines to avoid missing endocervical lesions.

Statistical analysis

The data analysis comprised descriptive statistics, with normally distributed continuous variables summarized as mean ± standard deviation, non-normally distributed variables presented as median and interquartile range (IQR), and categorical variables expressed as counts and percentages. Group comparisons used Welch’s t-test or ANOVA for continuous variables and Fisher’s exact test (expected frequencies < 5) or Chi-squared test for categorical variables. To identify factors associated with HSIL+, we first conducted univariate logistic regression analyses to evaluate the association between each variable and HSIL+. Variables with a p-value <0.05 in univariate analysis were subsequently included in the multivariate logistic regression model to adjust for potential confounders and determine independent predictors. Nonlinear relationships between Age and treatment outcomes were explored using restricted cubic splines (RCS) with knots at the 5th, 35th, 65th, and 95th percentiles. A global P value <0.05 indicated significance, and a P nonlinear <0.05 suggested nonlinearity. Breakpoints were identified using piecewise logistic regression, with model fit evaluated by the log-likelihood ratio test (P < 0.05 indicating better fit). In our study, all statistical analyses were performed using the R software (version 4.4.1; R Core Team, 2024).

Results

Characteristics of study subjects

The study included 4,648 women with positive cervical screening but no visible colposcopic findings (Fig. 1). Among 37,567 colposcopies, we excluded 10,577 Type 1/2 TZ cases and 597 post-total hysterectomy cases (including complete removal of the uterine corpus and cervix), leaving 26,393 TZ3 cases. Further exclusions included 7,761 with acetowhite lesions, 13,240 without histopathological confirmation, 389 without ECC, and 355 with incomplete data, leaving a final study population of 4,648 women. The baseline characteristics of the study population are summarized in Table 1. In this study, most participants were aged ≥50 years. Most patients had a history of 1–3 pregnancies or deliveries, and 24.1% tested positive for high-risk HPV types (HPV 16/18). Cytologies results revealed 36.0% with ASCUS and 15.2% with LSIL. The majority of patients underwent ECC in combination with multiple biopsy sites, with ECC + 2 biopsies being the most frequently performed. The detection rates of HSIL+ by biopsy and ECC were comparable, with 89 cases (2.1%) identified by biopsy and 91 cases (2.0%) by ECC, respectively. Collectively, these two modalities detected a total of 145 HSIL+ cases (3.1%).

Figure 1 Flowchart of data included.

TZ, Transformation Zone; ECC, endocervical curettage.

Table 1 Patient demographics and baseline characteristics.

Characteristic	N = 4,6481	
Age (year)		
<30	190 (4.1%)	
30–39	855 (18.4%)	
40–49	1,195 (25.7%)	
50–59	1,861 (40%)	
60–69	494 (10.6%)	
≥70	53 (1.1%)	
Gravidity		
0	181 (3.9%)	
1–3	2,767 (59.5%)	
<3	1,700 (36.6%)	
Parity		
0	335 (7.2%)	
1–3	4,236 (91.1%)	
<3	77 (1.7%)	
Menopause		
No	2,320 (49.9%)	
Yes	2,328 (50.1%)	
Cytology		
NILM	2,087 (44.9%)	
ASCUS	1,674 (36.0%)	
LSIL	705 (15.2%)	
ASC-H/HSIL/AGC	182 (3.9%)	
HPV		
Negative	286 (6.2%)	
Non-HPV 16/18	3,242 (69.8%)	
HPV 16/18	1,120 (24.1%)	
TestType		
ECC	602 (13.0%)	
ECC+1	478 (10.3%)	
ECC+2	1,842 (39.6%)	
ECC+3	782 (16.8%)	
ECC+4	944 (20.3%)	
Biopsy		
Negative	2,940 (68.9%)	
LSIL	1,238 (29.0%)	
HSIL+	89 (2.1%)	
ECC		
Negative	4,314 (92.8%)	
LSIL	243 (5.2%)	
HSIL+	91 (2.0%)	
Histologic diagnosis		
Negative	3,169 (68.2%)	
LSIL	1,334 (28.7%)	
HSIL+	145 (3.1%)	
Notes.

1 Median (IQR); n (%).

NILM negative for intraepithelial lesion or malignancy

ASCUS atypical squamous cells of undetermined significance

LSIL low-grade squamous intraepithelial lesions

HSIL high- grade squamous intraepithelial lesions

ASC-H atypical squamous cells that cannot exclude HSIL

AGC atypical glandular cells

HPV human papillomavirus

ECC endocervical curettage

Risk stratification of HSIL+ in women with no visible colposcopic findings based on cytology and HPV status

Women with abnormal cervical cancer screening results but no visible colposcopic findings present a diagnostic challenge. Our analysis (Table 2) revealed that ASC-H/HSIL/AGC on cytology was strongly associated with HSIL+ (OR = 22.57, P < 0.001), while ASCUS and LSIL did not significantly increase the risk. HPV 16/18 infection was significantly associated with HSIL+ (OR = 5.38, P = 0.002), and non-16/18 high-risk HPV types were also associated with an elevated risk (OR = 3.23, P = 0.029).

Table 2 Univariate and multivariate logistic regression analyses of clinical features associated with HSIL+ in women with colposcopy incomplete (transformation zone type 3) and without visible lesions.

Characteristic	Univariable	Multivariable	
	N	Event N	OR	95% CI	p-value	N	Event N	OR	95% CI	p-value	
Gravidity											
0	181	2	1.00	REF							
1–3	2,767	85	2.84	0.89, 17.32	0.147						
<3	1,700	56	3.05	0.94, 18.71	0.124						
Parity											
0	328	4	1.00	REF							
1–3	4,236	136	2.69	1.12, 8.78	0.053						
<3	77	3	3.28	0.64, 15.20	0.125						
Menopause											
No	2,320	70	1.00	REF							
Yes	2,328	73	1.04	0.75, 1.45	0.815						
Cytology											
NILM	2,087	44	1.00	REF		2,087	44	1.00	REF		
ASCUS	1,674	30	0.85	0.53, 1.35	0.488	1,674	30	0.96	0.59, 1.55	0.873	
LSIL	705	14	0.94	0.49, 1.68	0.844	705	14	1.11	0.58, 2.01	0.738	
ASC-H/HSIL/AGC	182	55	20.11	13.04, 31.20	<0.001	182	55	23.47	15.04, 36.91	<0.001	
HPV											
Negative	286	4	1.00	REF		286	4	1.00	REF		
Non-HPV 16/18	3,242	91	2.04	0.84, 6.69	0.167	3,242	91	3.28	1.30, 11.12	0.026	
HPV 16/18	1,120	48	3.16	1.27, 10.52	0.028	1,120	48	5.27	2.00, 18.24	0.002	
Notes.

Abbreviations CI Confidence Interval

OR Odds Ratio

Null deviance = 1,277; Null df = 4,647; Log-likelihood = −537; AIC = 1,086; BIC = 1,124; Deviance = 1,074; Residual df = 4,642; No. Obs. = 4,648

Interaction analysis showed no significant effect modification between the two variables (Tables S1 and S2), supporting the use of stratified rather than interaction-based modeling. This study stratified HSIL+ risk by cytology categories and HPV status (Table 3). High-grade cytology (ASC-H/HSIL/AGC) demonstrated high HSIL+ rates across all HPV subgroups (32.5%–37.2%), including a 4.5% risk in HPV-negative women. Regarding low-grade cytology (NILM/ASCUS/LSIL), HPV 16/18 positivity was associated with significantly elevated HSIL+ rates compared to non-16/18 HPV-positive cases, with risks ranging from 2.4% to 5.0% versus 1.6% to 1.8%, respectively. Notably, among women with LSIL cytology, the HSIL+ detection rate reached 5.0% in those with HPV 16/18.

Age-related risk of HSIL+ in women with NILM, ASCUS, or LSIL and non-HPV 16/18 positivity

Among 3,125 women with NILM, ASCUS, or LSIL cytology and non-HPV 16/18 positivity, 54 cases (1.73%) were diagnosed with HSIL+. Stratified by TCT category, HSIL+ was detected in 22 NILM cases, 23 ASCUS cases, and nine LSIL cases. RCS (Fig. 2) analysis demonstrated a nonlinear association between age and HSIL+ risk (P-nonlinear = 0.008), identifying a threshold at 55 years (Table 4). Further analysis by age group showed that among 2,123 patients under 55 years old, 32 (1.51%) had HSIL+, while among 1,002 patients aged 55 and older, 22 (2.20%) were HSIL+. While the standard logistic model showed no significant association between age and HSIL+ risk (adjusted OR = 1.01, P = 0.661), the piecewise model revealed distinct patterns: Below age 55, age had no significant impact on HSIL+ risk (OR = 0.96, P = 0.151). Among women aged 55 and older, each additional year of age was associated with a 10% increase in the odds of HSIL+ (OR = 1.10, 95% CI [1.02–1.19]; P = 0.015).The piecewise model significantly outperformed the standard model (log-likelihood ratio P = 0.022), with adjustments for gravidity, parity, menopause status, cytology results, and ECC/biopsy methods. To further illustrate this relationship, we performed age-stratified analysis in 5-year intervals (Table 5). The HSIL+ detection rates were 4.42% in women aged ≥65 years.

Figure 2 The association between age and HSIL+ using restricted cubic splines.

Age-related risk of HSIL+ in women with NILM, ASC-US, or LSIL and non-HPV 16/18 positivity. Model with four knots located at 5th, 35th, 65th and 95th percentiles. Y-axis represents the OR to present HSIL for any value of age compared to individuals with reference value (50th percentile) of age. The logistic regression was adjusted for gravidity, parity, menopause, TCT, and TestType.

Table 3 Correlation between cytology, HPV genotypes, and histopathologic HSIL+.

	TCT (NILM), N = 2087	TCT (ASCUS), N = 1674	TCT (LSIL), N = 705	TCT (ASC-H/HSIL/AGC), N = 182	
	HPV (-), N = 89	Non-HPV 16/18, N = 1,252	HPV 16/18, N = 746	HPV (-), N = 110	Non-HPV 16/18, N = 1,313	HPV 16/18, N = 251	HPV (-), N = 65	Non-HPV 16/18, N = 560	HPV 16/18, N = 80	HPV (-), N = 22	Non-HPV 16/18, N = 117	HPV 16/18, N = 43	
<HSIL	89 (100.0%)	1,230 (98.2%)	723 (96.9%)	108 (98.2%)	1,290 (98.2%)	245 (97.6%)	64 (98.5%)	551 (98.4%)	76 (95.0%)	21 (95.5%)	79 (67.5%)	27 (62.8%)	
HSIL+	0 (0.0%)	22 (1.8%)	23 (3.1%)	2 (1.8%)	23 (1.8%)	6 (2.4%)	1 (1.5%)	9 (1.6%)	4 (5.0%)	1 (4.5%)	38 (32.5%)	16 (37.2%)	
Notes.

Abbreviations HSIL+ high-grade squamous intraepithelial lesion or worse

NILM negative for intraepithelial lesion or malignancy

ASCUS atypical squamous cells of undetermined significance

LSIL low-grade squamous intraepithelial lesion

This table summarizes HSIL+ cases and detection rates by cytology and HPV genotype in women with type 3 transformation zone and no visible lesions. It demonstrates risk variation across subgroups to support individualized management.

Table 4 Threshold effect analysis of age on HSIL+.

	adjusted OR (95% CI)1	P-value	
Fitting by standard logistic regression model	1.01 (0.97, 1.06)	0.661	
Fitting by piecewise logistic regression model (Break-Point = 55)			
Age (year) < 55	0.96 (0.90, 1.02)	0.151	
Age (year) ≥ 55	1.10 (1.02, 1.19)	0.015	
Log likelihood ratio		0.022	
Notes.

1 Adjusted for: Gravidity, Parity, Menopause, Cytology, TestType.

The best performance is shown in bold

Table 5 Age-stratified HSIL+ detection rate among women aged ≥55 years with HPV Non-16/18 and low-grade cytology.

Age group	Total	HSIL+	HSIL+ (%)	
55–59	572	12	2.10	
60–64	197	4	2.03	
≥65	113	5	4.42	
Notes.

Age-stratified analysis among women aged ≥ 55 years with HPV non-16/18 infection and low- grade cytology revealed increasing HSIL+ detection rates with age. The rates were 4.42% in ≥ 65.

Stratified analysis of the additional diagnostic yield of cervical biopsy in HSIL+ detection

According to ASCCP guidelines, a ≥4% risk of CIN3+ warrants diagnostic intervention. In our study, although the endpoint was HSIL+ (CIN2+), we conservatively applied the same 4% threshold to define high-risk subgroups, given the diagnostic challenges in TZ3 cases with no visible lesions.For the high-risk subgroups identified in our study, we further analyzed patients who underwent both ECC and cervical biopsy to evaluate the additional diagnostic value of biopsy beyond ECC alone (Table 6).

Patients with cervical cytology = ASC-H/HSIL/AGC (n = 182): ECC detected 23.1% of HSIL+ cases, with cervical biopsy adding an additional 7.1%. In HPV 16/18 positive patients, biopsy significantly increased detection by 11.6%, while in non-HPV 16/18 cases, it contributed 6.8%.

Patients with cervical cytology = LSIL and HPV 16/18 positive (n = 73): four HSIL+ cases were detected, with two cases (2.7%) identified by ECC and an additional 2 (2.7%) by biopsy, resulting in an incremental yield of 2.7% (95% CI: 0.3%–9.5%).

Table 6 Incremental diagnostic yield of cervical biopsy for HSIL+ detection.

	Total HSIL+ Cases (n)	HSIL+ Detected by ECC (n (%))	Additional HSIL+ Detected by Biopsies1 [n (%)]	Incremental Yield (% (95% CI))2	
Cytology= ASC-H/HSIL/AGC(n = 182)	
HPV=all(n = 182)	55	42 (23.1)	13 (7.1)	7.1 (3.9–11.9)	
HPV=Negative(n = 22)	1	1 (9.1)	0 (0.0)	0.0 (0.0–15.4)	
HPV=Non-HPV 16/18(n = 117)	38	30 (25.6)	8 (6.8)	6.8 (3.0–13)	
HPV=HPV 16/18(n = 43)	16	11 (25.6)	5 (11.6)	11.6 (3.9–25.1)	
Cytology= LSIL, HPV=HPV16/18(n = 73)					
Cytology=LSIL (n = 73)	4	2 (2.7)	2 (2.7)	2.7 (0.3–9.5)	
Cytology= NILM/ ASCUS/LSIL,
HPV=Non-HPV 16/18,
Age≥65(n = 113)					
Cytology=all(n = 113)	5	2 (1.8)	3 (2.7)	2.7 (0.6–7.6)	
Cytology=NILM(n = 49)	3	1 (2.0)	2 (4.1)	4.1 (0.5–14.0)	
Cytology=ASCUS(n = 47)	1	0 (0.0)	1 (2.1)	2.1 (0.1–11.3)	
Cytology=LSIL(n = 17)	1	1 (5.9)	0 (0.0)	0.0 (0.0–19.5)	
Notes.

1 Additional HSIL+ Detected by Biopsies refers to cases detected through biopsies but missed by ECC.

2 Data presented as n (%) with 95% confidence intervals.

Patients with cervical cytology = NILM/ASCUS/LSIL, HPV non-16/18, Age ≥65 (n = 113):biopsy contributed additional diagnostic value. Among five HSIL+ cases, two were detected by ECC and three were identified only by biopsy, yielding an incremental detection rate of 2.7%. Stratified by cytology, the incremental biopsy yield was 4.1% in NILM cases, 2.1% in ASCUS, and 0% in LSIL, where ECC alone detected the single HSIL+ case.

To further illustrate the complementarity of ECC and biopsy, we cross-classified their HSIL+ detection (Table S3). Among 4,648 women, 34 cases were positive on both ECC and biopsy, 57 cases were identified only by ECC, and 53 cases only by biopsy, while 4,504 were negative on both. These findings demonstrate that ECC and biopsy each contributed unique diagnostic value in women with TZ3 and no visible colposcopic findings.

Our retrospective study identified five patients with abnormal cervical screening results who had no visible lesions on colposcopy but were subsequently diagnosed with invasive cervical cancer through pathological examination (Table S4). All patients had a history of high-risk HPV infection, regardless of HPV type or cervical cytology results (ranging from NILM to ASC-H). Among them, three patients had documented persistent high-risk HPV positivity, while the duration was unknown in two cases. Two patients had a history of cervical intraepithelial neoplasia (CIN) and underwent conization. Notably, two patients were in their 30s, and three patients were in their 40s. ECC identified cervical squamous cell carcinoma in two cases (Cases 1 and 2) and adenocarcinoma in one case (Case 3), while biopsies revealed squamous cell carcinoma in two other cases (Cases 4 and 5).

Discussion

Our study focused on women with abnormal cervical screening and a type 3 transformation zone without visible lesions, in whom lesion detection is limited. We found that HSIL+ risk varied by cytology, HPV genotype, and age. High-grade cytology was strongly associated with HSIL+, supporting immediate diagnostic intervention. Among women with low-grade cytology, those with HPV 16/18—especially with LSIL—had HSIL+ rates exceeding 4%, meeting the ASCCP threshold for colposcopy and further evaluation. In contrast, women under 65 years with low-grade cytology and non-16/18 HPV had low risk and may be managed conservatively. Notably, in women aged ≥65 with low-grade cytology and non-16/18 HPV, the HSIL+ detection rate over the 4% threshold commonly used to guide immediate intervention. These findings highlight the value of incorporating age into risk stratification for this challenging subgroup.

A pooled analysis of 11 population-based cervical cancer screening studies (Zhao et al., 2020) included 3,317 women with abnormal screening results but no visible colposcopic findings at baseline. All participants underwent four-quadrant random cervical biopsies, revealing 177 cases of CIN2+, resulting in a detection rate of 5.3% (177/3,317). In our study, among 4,648 women with abnormal cervical screening results but no visible lesions at initial colposcopy, 3.1% (145 cases) were subsequently diagnosed with occult high-grade lesions.

It is well-established that cervical cytology findings of ASC-H/HSIL/AGC and HPV 16/18 positivity are risk factors for HSIL+ (Perkins et al., 2023; Rodríguez-Trujillo et al., 2018). In our study, we further clarified the risk stratification across different subgroups. To explore whether the effect of HPV genotype varied by cytology category, we examined their interaction using a Firth logistic regression model. Both were independent predictors of HSIL+, but no significant interaction was found, suggesting that their effects on HSIL+ risk are separate and do not modify each other.For high-grade cytology (ASC-H/HSIL/AGC), the risk of HSIL+ remained consistently high (32.5%–37.2%) across all HPV subgroups, including a 4.5% risk even in HPV-negative women. This highlights the necessity of immediate colposcopy and further diagnostic evaluation for these cases, regardless of HPV status. On the other hand, for low-grade cytology (NILM/ASCUS/LSIL), HPV 16/18 positivity was associated with a significantly higher risk of HSIL+ (2.4%–5.0%) compared to non-16/18 HPV-positive cases (1.6%–1.8%). Notably, the highest HSIL+ detection rate in this group was observed in women with LSIL cytology and HPV 16/18 infection (5.0%). These findings underscore the importance of differentiating cytologic severity when interpreting HPV results. In particular, LSIL cases with concurrent HPV 16/18 positivity may warrant immediate colposcopy and further diagnostic evaluation, given the elevated risk of underlying high-grade lesions.

Furthermore, our study specifically focused on women who were HPV non-16/18 positive with low-grade cytology findings (NILM, ASCUS, or LSIL) and no visible colposcopic lesions. Among this group, the HSIL+ rate was 1.73%, which, although relatively low, remains clinically significant. Additionally, we conducted RCS analysis to explore the relationship between age and HSIL+ risk. The analysis revealed that 55 years of age served as a threshold. For women older than 55, the risk of HSIL+ was 2.20%, and each additional year of age was associated with a 10% increase in the risk of HSIL+. To further illustrate this relationship, we performed age-stratified analysis in 5-year intervals. The HSIL+ detection rates were 4.42% in women aged ≥65 years. This supports the regression-based finding of a steady age-related increase in HSIL+ risk.

Although the overall HSIL+ rate in this population was relatively low (1.73%), which may not exceed the referral threshold set by risk-based guidelines ASCCP (≥4% for CIN3+) (Cheung et al., 2020), the significantly higher rates observed in older age groups suggest that age may be an important risk modifier. Moreover, all women in this study had a type 3 transformation zone, which is associated with a higher likelihood of missed lesions due to non-visible SCJ. In such cases, particularly in women ≥65 with persistent HPV infection and limited follow-up capacity, colposcopy combined with ECC may be clinically justifiable despite sub-threshold risk levels. These findings support more individualized, age-informed management strategies in this subgroup.

In women with type 3 transformation zones and no visible colposcopic findings, where lesion visualization is frequently limited due to an invisible squamocolumnar junction, the accuracy of cervical biopsy may be compromised by the absence of visible target areas. In this context, ECC plays a pivotal role in assessing potential endocervical lesions that may otherwise be missed. Current US guidelines recommend ECC when the SCJ is not fully visible (Massad et al., 2023; Wright et al., 2007). Given that our study exclusively included patients with type 3 transformation zones, ECC was routinely performed. To further evaluate the significance of cervical biopsy, we analyzed its additional detection rate beyond ECC. In patients with high-grade cytology (ASC-H/HSIL/AGC), both ECC and biopsy showed high detection rates, with biopsy adding an additional 7.1% yield, supporting their combined use for comprehensive evaluation. In women with LSIL cytology and HPV 16/18 infection, biopsy alone detected two of four HSIL+ cases, contributing an incremental yield of 2.7%. Likewise, in women aged ≥65 with low-grade cytology and non-HPV 16/18 infection, biopsy identified three out of five HSIL+ cases, also with an incremental yield of 2.7%. Stratified analysis indicated the highest biopsy benefit in NILM cases (4.1%), followed by ASCUS (2.1%). Given this substantial improvement, we recommend routine cervical biopsy in addition to ECC for this subgroup. Hu et al. (2017) similarly demonstrated that random biopsies plus ECC are essential for ASC-US/LSIL patients with any high-risk HPV(hrHPV) infection (including non-16/18 types), showing significantly increased detection of CIN2+ (OR: 4.1, 95% CI [2.6–6.4])and CIN3+ (OR: 6.5, 95% CI [2.5–17.3]).

These five cases of invasive cervical cancer all had a history of high-risk HPV infection, whether HPV 16/18 or non-16/18. While the duration of HPV persistence was unknown in two patients, three had documented persistent high-risk HPV positivity. Two patients also had a history of CIN. Notably, relying solely on ECC or biopsies may result in missed diagnoses. The key clinical insight is that persistent hrHPV infection (≥1 year) should be considered a significant risk factor, especially in patients with a history of CIN, even when colposcopy fails to detect visible lesions. Colposcopists should prioritize these patients for comprehensive evaluation, combining ECC and biopsies, to ensure accurate diagnosis and timely intervention. This approach reduces the risk of missed diagnoses and improves outcomes for women with persistent hrHPV infections.

This study has several important limitations that should be considered. First, the retrospective, single-center design may introduce selection bias and limit generalizability. Second, while both ECC and biopsies were performed, the anatomical limitations of type 3 transformation zones—specifically the non-visible squamocolumnar junction—may have hindered the collection of representative tissue, leading to missed lesions and potential underestimation of HSIL+ risk. Third, we did not examine how different non-16/18 HPV genotypes or co-infections affect HSIL+ risk. Fourth, although we explored the interaction between cytology and HPV genotype, the statistical power may have been limited in some subgroups due to small event numbers. Thus, the absence of significant interactions should be interpreted with caution. Another important limitation is TZ classification may be partly subjective. Although two independent experts re-reviewed the images with results consistent with the original classification, potential misclassification cannot be fully excluded and should be considered when interpreting the results. Finally, our analysis adopted the ≥4% ASCCP threshold for CIN3+ to define high-risk subgroups, although our endpoint was HSIL+ (CIN2+). This conservative approach minimized underestimation in TZ3 women but may not fully reflect the lower malignant potential of CIN2. A higher threshold (∼6%) could be more appropriate and requires future validation.

Conclusion

This study highlights the value of risk-based management in women with abnormal cervical screening but no visible colposcopic findings, particularly in those with a type 3 transformation zone. According to the 2019 ASCCP guidelines (Perkins et al., 2020), an immediate risk of ≥4% for CIN3+ supports diagnostic intervention. Our results suggest that this threshold is exceeded in several key subgroups, including women with high-grade cytology, those with HPV 16/18 and LSIL cytology, and women aged ≥65 years with low-grade cytology and non-16/18 HPV. For these groups, immediate diagnostic procedures such as ECC and biopsy may be warranted. In contrast, women with HPV 16/18 and NILM or ASCUS cytology, as well as those with non-16/18 HPV, low-grade cytology, and age <65 years, showed relatively low HSIL+ risk and may be considered for conservative follow-up when appropriate.

Supplemental Information

Supplemental Information 1 Raw data for Table 1–Table 3

Supplemental Information 2 Raw data for Table 4 and Fig. 2

Supplemental Information 3 Raw data for Table 5

Supplemental Information 4 Raw data of patients with cervical cytology = LSIL and HPV 16/18 positive (n = 73)

Supplemental Information 5 Raw data of patients with cervical cytology = NILM/ASCUS/LSIL, HPV non-16/18, Age ≥ 65 (n = 113)

Supplemental Information 6 Raw dataset used for the analysis in Table 5

Supplemental Information 7 Firth Penalized Logistic Regression Results with Cytology and HPV Interaction Terms for Predicting HSIL+

Supplemental Information 8 P-values for Main and Interaction Effects in Firth Logistic Regression Model Predicting HSIL+

Supplemental Information 9 Cross-classification of HSIL+ detection by ECC and cervical biopsy in women with type 3 transformation zone and negative colposcopy (N = 4, 648)

Supplemental Information 10 Characteristics of 5 patients diagnosed with invasive cervical cancer

CIN, cervical intraepithelial neoplasia.

Supplemental Information 11 STROBE Checklist for Cross-Sectional Study

We extend our gratitude to the staff at the Cervical Disease Center for providing the data essential to this study.During the preparation of this manuscript, the authors utilized ChatGPT exclusively for English language polishing, including grammar checking and academic expression refinement. All AI-generated content was rigorously reviewed and modified by the authors. No AI tools were employed in any aspect of research design, data analysis, or interpretation of results. The authors take full responsibility for the intellectual content and scientific integrity of this work.

Additional Information and Declarations

Competing Interests

Author Contributions

Ethics

Data Availability

The authors declare there are no competing interests.

Jing Yang conceived and designed the experiments, performed the experiments, analyzed the data, prepared figures and/or tables, and approved the final draft.

Qiao Liu conceived and designed the experiments, performed the experiments, authored or reviewed drafts of the article, and approved the final draft.

Yi Tang performed the experiments, analyzed the data, authored or reviewed drafts of the article, and approved the final draft.

Kui Huang analyzed the data, prepared figures and/or tables, and approved the final draft.

Tianmin Chen performed the experiments, authored or reviewed drafts of the article, and approved the final draft.

Jing Zhao conceived and designed the experiments, authored or reviewed drafts of the article, and approved the final draft.

The following information was supplied relating to ethical approvals (i.e., approving body and any reference numbers):

This research was approved by the Ethics Committee of Hunan Provincial Maternal and Child Health Care Hospital (Approval No. 2023-S180).

The following information was supplied regarding data availability:

The raw measurements are available in the Supplemental Files.

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
