# Peer review of "Retrospective analysis of cervical screening abnormalities in women with type 3 transformation zone without visible lesions"

_PeerJ, doi:10.7717/peerj.20396_

## Round 0.1 · original submission · Major Revisions

· Academic Editor

Major Revisions

Reviewer 1 ·

Basic reporting

Thank you for the opportunity to review this paper. The authors present a large data set of colposcopic data which is to be commmended.
The article is well written.

In the abstract I would recommend some background as to why this is a clinically relevant / contentious issue.

As a reader, it would be useful for me to know how primary cervical screening works in the region studied. Is this HPV testing with reflex cytology or cytology with reflex HPV or does everybody get both HPV testing & cytology and anyone with an abnormality in either is referred for colposcopy. Different jurisdictions have different screening guidelines, and the context of the local guidelines would be helpful for interpreting the article.

Experimental design

Your sample group is women with abnormal cervical screening, a normal colposcopy and type 3 transformation zone.

Two thirds of your population having a type 3 TZ (26393 / 37567 colposcopies) sounds very high. This contrasts with this paper which reports 10-15 % of women presenting for screening having a Type 3 TZ. Manga SM, Kincaid KD, Boitano TKL, Tita AT, Scarinci IC, Huh WK, Liang MI. Misoprostol and estradiol to enhance visualization of the transformation zone during cervical cancer screening: An integrative review. Eur J Obstet Gynecol Reprod Biol. 2022 Feb;269:16-23. doi: 10.1016/j.ejogrb.2021.11.431. Epub 2021 Nov 30. PMID: 34952401; PMCID: PMC10958763.
This discrepancy concerns me about the generalizability of the results.

Clinically, how do you manage people with HG cytology and a negative colposcopy and biopsies. In my own clinical practice, these cases would be reviewed in a multidisciplinary meeting. Women with confirmed HG cytology and negative biopsies & colposcopies would likely undergo a diagnostic Lletz procedure. I am not clear how you managed these cases.

My key concern regarding this study is that I am not convinced you have identified everyone with a HG lesion - it is possible there was a negative colposcopy (HG lesion missed), negative biopsies (wrong place biopsied) and a HG lesion still present. This could potentially be addressed by longer term follow up, or diagnostic excision procedures for those with HG cytology.

Validity of the findings

The link with HG cytology and high risk HPV subtype with HG histology is anticipated.

As stated above, my concern as to the validity of these findings is how you can be certain (you can't) that all HG lesions are detected. This is acknowledged in your discussion but I think it is a key weakness in the study.

Additional comments

Table 4
1. "confidence" is spelled incorrectly - typo.

·

Basic reporting

See below

Experimental design

See below

Validity of the findings

See below

Additional comments

Transformation zone type 3 is a great clinical challenge for clinical colposcopists. Cervical screening programs were originally designed for women in fertile ages with visible transformations zones and visible, accessible lesions. Women with TZ3 are less suitable targets for screening but still benefit from it. They face higher risks for excessive unbeneficial and uncomfortable interventions and both over- and undertreatment. And colposcopists constantly struggle with the problem – what do we do when we cannot see a lesion and cannot rule out there is one?

Yang and coworkers help us out with a large study that can be used for evaluation of parameters at hand, when we cannot see the most vulnerable area. The material is complete after excluding patients with missing data. It is almost inconceivable that only 7% (n=355) had any missing data after collecting 8 different variables from each patient!

Strength: The strength of the study is that all women with either HPV or any degree of cytological abnormality went through colposcopy, biopsies and ECC. This allows for calculation of risks for all combinations of cytology and HPV-status. Most important findings, in this reviewer’s view, are not the expected high risks for high grade cytology, but the very low risks shown, <3%, for Cytology ≤ ASCUS and LSIL/HPV non16/18, that allows a much more relaxed follow up with benefits for the patients and the health care system.

Limitations/concerns: The conclusion is not valid and not directly derived from the results. The data cannot be converted into guidelines automatically. Actions and follow-up is dependent how the screening program works. The authors should note that their experimental setting is unreasonable in the real world – referral to colposcopy need some kind of triage and cannot be based on only one ASCUS or a single test with HPV non16/18 and normal cytology.

In my view this is not a cohort study but a cross-sectional study and the headline should be changed accordingly.

The authors refer to risk-based management and in several places comparisons between groups of women with very low risks are compared. Thresholds can vary, but I suggest that the authors refer to US guidelines, where the threshold for colposcopy is a ≥4% risk for CIN3+. This could be translated to a higher threshold when CIN2 is included. (E.g. Canadian guidelines with ≥5%). In countries with lower resources than these rich countries thresholds might need to be adjusted even more upwards.

Histology is gold standard in this study, and that is the only possible choice. Caution should be noted and expressed, however, as TZ3, the state studied, makes it difficult to obtain tissue representative for a possible lesion. Risks can be underestimated.

The authors should explain the population and why among 37567 women, only 11174 had TZ 1 or 2. 70% of the population had TZ3!? This might need a short description of the screening program in the catchment area. The age distribution is not accounted for. Table 1 includes a number (50) for age that is not explained. Please include data in strata (i.e. <30, 30-40, 40-50 etc)

The value of cervical curettage is debated and the method is often not used in Europe, where it is replaced by endocervical cytology or diagnostic excision. It would be of value if the authors also could analyse data for biopsies and investigate the additional value of ECC, not only the other way around. This could be done by adding one or two columns to Table 5.

Cytology and HPV-type are treated as independent variables which is highly unlikely. They most probably are linked. The association should be explored and a possible interaction term entered in the analysis.



More specific comments:
L65 I suggest not restricting this to Thinprep®, or not even to liquid cytology. Well organised screening programs were very successful also with classical PAP-smears.
L97 No, long term risks are not assessed. This is a cross sectional study and that should be stated. Also incorrect description in abstract
L128 Claiming that informed consent is not needed in retrospective studies is not obvious, and incorrect in the tradition of Western countries relying on the Helsinki declaration. It might be acceptable to refrain from informed consent, however, if the researchers can show how the integrity of the patients can be preserved. This could be a matter of anonymisation of personal data and security and integrity of data management should be declared.
L130 I suggest that the filled-in STORBE check list is provided as a supplement
L154 Liesegang does not provide a DSI colposcope
L158 Explain “necessary”
L168 Mosaicism and punctuation can only appear in acetowhite epithelium
L179 How were discrepancies resolved?
L185 Could 16 biopsies have been taken? Or is “One to four” the total number of biopsies?
L220 Why were variables considered confounders despite not reaching significant association with outcome?
L220 Total hysterectomies?
L229 I suggest using the term cytology/cytologies throughout the manuscript and tables, instead of “TCT” which is not an established term
L247 To justify the term “major predictor” HPV16/18 should be the most common variable among CIN2+, which requires other comparisons. Please rephrase.
L337 Discussion. The first part could be moved to the introduction, and Discussion preferably start with highlighting the most important findings. Some limitations are listed but the strengths of the study could be summarized more clearly and the discussion could be more structured.
L384 To further prove and illustrate this point I suggest adding stratified age (see above) with proportions of HSIL+ to the table. Maybe 5-year strata should be used. This should prove their calculation that 65 year old women has twice the risk for HSIL+ as 55 year old women (10*10% increase per year), if this is really the case.
L399 As US guidelines are referred to, I recommend this should be written out
L417 According to what guideline should patients with a 97.6% probability of not having HSIL+ undergo colposcopy and curettage at all? (see above)
L457 The study in itself does not establish av management strategy. It can be the basis for a strategy or enable such strategy. A strategy can be recommended but external validity of such recommendation is limited and dependent of the setting
L465 I would consider this a misinterpretation of the data. The effect of age is low. The data don’t show annual increase (by calender year), but increase in age per year gained. In the analysis adjustments have been made for cytology and HPV-results. Age could be considered in management, but I don’t find the data that show this should be regardless of cytology finding. Consider also that if one age group have higher risk and should qualify for more intense interventions, other age groups should have more relaxed management.

Table 2
The headline uses the term Negative colposcopy. This is not correct. I suggest: Colposcopy incomplete (Transformation zone type 3) and without visible lesions.
Please clarify reference values (“1” is better than “–“). Write out that all variables were included in the multivariat analysis.

Table 3
This is the most important table in the manuscript. Please make the description of the table more complete (should be possible to understand without reading the whole manuscript).

---

## Round 0.2 · Major Revisions

· Academic Editor

Major Revisions

Reviewer 1 ·

Basic reporting

Good.
Line 81 'how to manage women with' doesn't read well. Suggest 'the best way to manage..'
Line 340 "fourth' should be 'fourth'.

Experimental design

In the abstract, you comment on the number with HG+. I think you could still be clearer - HG+ identified; i.e, there will be some people with HG+ who you haven't identified.

Validity of the findings

-

Additional comments

Thank you for the adjustments and modifications made.
An interesting paper and some food for thought in this clinically challenging area.

·

Basic reporting

-

Experimental design

-

Validity of the findings

-

Additional comments

This manuscript has been significantly improved, and I am happy to note the changes made by the authors. However, there are still one major and some minor issues that should be resolved before I would recommend publishing the paper.

MAJOR
Both reviewers have reacted to the very high proportion of patients with Transformation zone type 3 in the study base. The authors’ explanations are not satisfactory. They don’t explain why their population (the 37567 women) should be so much different from a normal colposcopy population within a screening program. As they investigate all test-positive patients without any kind of triage, the average age of the women referred to colposcopy should be lower. The matter is important as it opens up for a potential misclassification. If a large proportion of those women classified as TZ 3 actually have TZ 2 or 1, this would bias the results, increasing the probability of finding HSIL+ with biopsies and even cervical curettage. Furthermore, it would reduce the validity of the study as it should be based on a TZ 3 population. It would also question the competence of the colposcopists involved, not being able to visualize a squamous columnar junction that should be within reach. The possibility of misclassification should at least be discussed as a major limitation in the discussion part.

MINOR
Line 22 Population consists of women with TZ 3 is not mentioned
Line 37 and other parts of the manuscript. The authors agreed not to use the term negative colposcopy for incomplete/unsatisfactory colposcopies, but still do it, repeatedly. Colposcopy can only be negative if the assessment is complete. If incomplete, the phrase used should be Without visible lesions
L 101 Please add a reference
L 261 The number of participants remaining after exclusions (4648) should be in this paragraph.
L345 The authors argue against showing correlation between HSIL+-findings by cervical curettage and biopsies in women with TZ3 and no visible lesions, as cc is the “primary” investigation. However, the number of patients undergoing biopsies is almost the same. I would have appreciated a simple table. Two-by-two = Four cells: CC neg for HSIL+/Biopsy neg for HSIL+, CC pos/biopsy neg, CC neg/biopsy pos, and CC pos/biopsy pos. Then the reader could draw his/her own conclusion about the importance of performing either investigation. This would enhance the interest and external validity of the findings.
L411 and L597. This is not correct. This study includes CIN2, inferring a lower risk for cancer development. A suitable threshold should be adjusted accordingly. (possibly ≈ 6%)
L483 – 492 Good!

---

## Round 0.3 · accepted · Accept

· Academic Editor

Accept

Thank you for revising your manuscript to address the concerns of the reviewers. Although reviewer 2 offers an optional suggestion to enhance supplementary table 3, the reviewers and I are otherwise satisfied with the revisions you have made in response to the various comments. The manuscript is now ready for acceptance and any minor change to supplementary table 3 can be handled in proof.

·

Basic reporting

Please see comment below

Experimental design

Please see comment below

Validity of the findings

Please see comment below

Additional comments

I am pleased about the authors found reviewers' earlier comments fruitful and have revised the manuscript accordingly. The manuscript is satisfactory revised and I recommend it to be published.

A very minor comment: In the supplementary table 3 it would have been nice to have No biopsies and Biopsies negative for HSIL+, separated. It would have been more informative. This should not be a reason to postpone publication, but perhaps this small revision could be a part of the copyproof process.